

# Nutrient uplift in a cyclonic eddy increases diversity, primary productivity and iron demand of microbial communities relative to a western boundary current

Martina A. Doblin[1], Katherina Petrou[2], Sutinee Sinutok[1,3], Justin R. Seymour[1], Lauren F. Messer[1], Mark V. Brown[4], Louiza Norman[5], Jason D. Everett[6], Allison S. McInnes[1], Peter J. Ralph[1], Peter A. Thompson[7] and Christel S. Hassler[8]

[1] Plant Functional Biology and Climate Change Cluster, University of Technology Sydney, Ultimo NSW, Australia
[2] School of Life Sciences, University of Technology Sydney, Ultimo NSW, Australia
[3] Faculty of Environmental Management, Prince of Songkla University, Kho Hong Songkhla, Thailand
[4] School of Biotechnology and Biomolecular Sciences, University of New South Wales, Sydney NSW, Australia
[5] Department of Plant Sciences, University of Cambridge, Cambridge, United Kingdom
[6] School of Biological, Earth and Environmental Sciences, University of New South Wales, Sydney NSW, Australia
[7] Oceans and Atmosphere Flagship, Commonwealth Scientific Industrial Research Organisation, Hobart Tas, Australia
[8] Institute F.-A. Forel, Earth and Environmental Sciences, University of Geneva, Geneva, Switzerland

Corresponding author
Martina A. Doblin,
Martina.Doblin@uts.edu.au

## ABSTRACT

The intensification of western boundary currents in the global ocean will potentially influence meso-scale eddy generation, and redistribute microbes and their associated ecological and biogeochemical functions. To understand eddy-induced changes in microbial community composition as well as how they control growth, we targeted the East Australian Current (EAC) region to sample microbes in a cyclonic (cold-core) eddy (CCE) and the adjacent EAC. Phototrophic and diazotrophic microbes were more diverse (2–10 times greater Shannon index) in the CCE relative to the EAC, and the cell size distribution in the CCE was dominated (67%) by larger micro-plankton ($\geq 20\mu m$), as opposed to pico- and nano-sized cells in the EAC. Nutrient addition experiments determined that nitrogen was the principal nutrient limiting growth in the EAC, while iron was a secondary limiting nutrient in the CCE. Among the diazotrophic community, heterotrophic *NifH* gene sequences dominated in the EAC and were attributable to members of the gamma-, beta-, and delta-proteobacteria, while the CCE contained both phototrophic and heterotrophic diazotrophs, including *Trichodesmium*, UCYN-A and gamma-proteobacteria. Daily sampling of incubation bottles following nutrient amendment captured a cascade of effects at the cellular, population and community level, indicating taxon-specific differences in the speed of response of microbes to nutrient supply. Nitrogen addition to the CCE community increased picoeukaryote chlorophyll *a* quotas within 24 h, suggesting that nutrient uplift by eddies causes a 'greening' effect as well as an increase in phytoplankton biomass. After three days in

both the EAC and CCE, diatoms increased in abundance with macronutrient (N, P, Si) and iron amendment, whereas haptophytes and phototrophic dinoflagellates declined. Our results indicate that cyclonic eddies increase delivery of nitrogen to the upper ocean to potentially mitigate the negative consequences of increased stratification due to ocean warming, but also increase the biological demand for iron that is necessary to sustain the growth of large-celled phototrophs and potentially support the diversity of diazotrophs over longer time-scales.

## INTRODUCTION

There are two broad nutrient limitation regimes for phytoplankton growth in the contemporary ocean, whereby iron (Fe) limitation occurs across ∼30% of the ocean's surface area where high macronutrient concentrations occur (high latitudes, upwelling and some coastal areas), and nitrogen (N) limitation occurs across most of the oligotrophic low-latitude systems (*Moore et al., 2013*). Different phytoplankton groups can have specific nutrient requirements, such as Fe for diazotrophs (*Kustka et al., 2003*) and silicon (Si) for diatoms (*Brzezinski & Nelson, 1989*), which may lead to secondary or interactive effects on Fe or N limitation. Co-limitation may also arise when there is physical mixing between water masses with different nutrient stoichiometry, or over seasonal cycles when physical nutrient inputs and biological cycling alters nutrient bioavailability (*Deutsch & Weber, 2012*).

Global Climate Model (GCM) projections indicate warming and increased stratification of the upper ocean over the coming decades, limiting the upwards delivery of nitrogen (e.g., "new" N) into the euphotic zone, and potentially leading to increased reliance on a smaller pool of regenerated forms or N fixation to support primary production (*Behrenfeld, 2011*). However, these models typically do not consider the influence of smaller scale oceanographic features such as meso-scale eddies, which could act as a compensatory mechanism and enrich the upper ocean with new nutrients delivered from deeper ocean waters, potentially mitigating the negative consequences of climate change (*Matear et al., 2013*).

While eddies are universal features of the global ocean (*Chelton, Schlax & Samelson, 2011*), they differ in their mode of formation, direction of rotation, size, longevity, and processes driving nutrient dynamics (e.g., interaction between wind and surface currents, horizontal entrainment; *Gaube et al., 2014*), and can thus have different biological effects (*Bibby et al., 2008*). Eddies formed in coastal regions, for example, can entrain enriched continental shelf water (*Waite et al., 2007*), and consequently have higher positive chlorophyll-a anomalies than their oceanic counterparts (*Everett et al., 2012*). The ratio of upwelled nutrients (e.g., Si:N) delivered into the euphotic zone is also important in determining the structure of microbial communities sustained by eddies and influences their biogeochemical function (*Bibby & Moore, 2011*). Therefore, the role eddies play in regulating internal nutrient inputs from the deep ocean is likely to be regionally dependent.

Much of what is known about eddies and their impact on the base of the foodweb is derived from satellite ocean colour observations which provide limited information with respect to microbial species composition and biogeochemical activity, and have a restricted view of the upper ocean (*McGillicuddy, 2016*). *In situ* observations and manipulative experiments are therefore critical to develop a full understanding of the role of meso-scale eddies in upper-ocean ecosystem dynamics and biogeochemical cycling.

Australia has one of the longest north-south coastlines in the world, and the oceanography along its east coast is extremely dynamic. It is strongly influenced by the flow of the Eastern Australian Current (EAC), one of five western boundary currents (WBCs) in the global ocean. Southward of ∼32°S 153°E, where two thirds of the EAC deviates eastward towards New Zealand to form the Tasman Front, the remaining EAC flow meanders, breaking down into a complex series of meso-scale eddies (*Ridgway & Godfrey, 1997*). The number and frequency of these eddies is higher than in the broader Tasman Sea, and their biological properties differentiate more strongly from background ocean waters than their oceanic counterparts (*Everett et al., 2014*). A critical research question is whether the observed intensification of the EAC (*Wu et al., 2012*) will result in more eddies, a change in the nature of these eddies, and how they will impact on primary producers and higher trophic levels.

Given that offshore phytoplankton communities in the Tasman Sea are generally N limited (*Hassler et al., 2011*; *Ellwood et al., 2013*), and that cyclonic cold-core eddies displace isopycnal surfaces (seawater of similar density) upwards and can upwell nutrients into the euphotic zone (*McGillicuddy et al., 1998*), EAC-induced eddies could increase the supply of nitrogen into surface waters from below the thermocline and potentially alter controls on phytoplankton growth. However, another source of N into surface waters is from $N_2$ fixing microbes. At least four different groups of diazotrophs have been identified from Australian waters: the filamentous, photosynthetic cyanobacterium *Trichodesmium*, the unicellular phycoerythrin-containing cyanobacterium *Crocosphaera watsonii* (group B), the photoheterotrophic symbiont UCYN-A (associated with specific prymnesiophyte hosts; *Hagino et al., 2013*), and heterotrophic proteobacteria (*Moisander et al., 2010*; *Seymour et al., 2012*; *Messer et al., 2015*). The regional distribution of diazotrophs in the western South Pacific suggests there is a sufficient supply of Fe to satisfy the requirements of the nitrogenase enzyme (*Kustka et al., 2003*), but little is known about how eddies may change the delivery of Fe or N from depth in this region, and hence alter the dynamics of diazotrophs relative to other microbial groups.

This study targeted the western Tasman Sea, which is dominated by the East Australian Current (EAC) and its associated eddy field. We examined the composition and diversity of the microbial community in a cyclonic cold core eddy (CCE) and in the EAC, and examined their responses to separate additions of nitrate (N), nitrate with iron (NFe), silicic acid (Si) as well as a mix of nutrients containing N, Fe, Si and phosphate (P). We focused on communities from the subsurface chlorophyll-a maximum to simulate the impact of moderate nutrient uplift into the euphotic zone (i.e., upwelling that does not reach the surface), and advance knowledge about responses of microbial communities that are difficult to detect using satellites.

Peerj

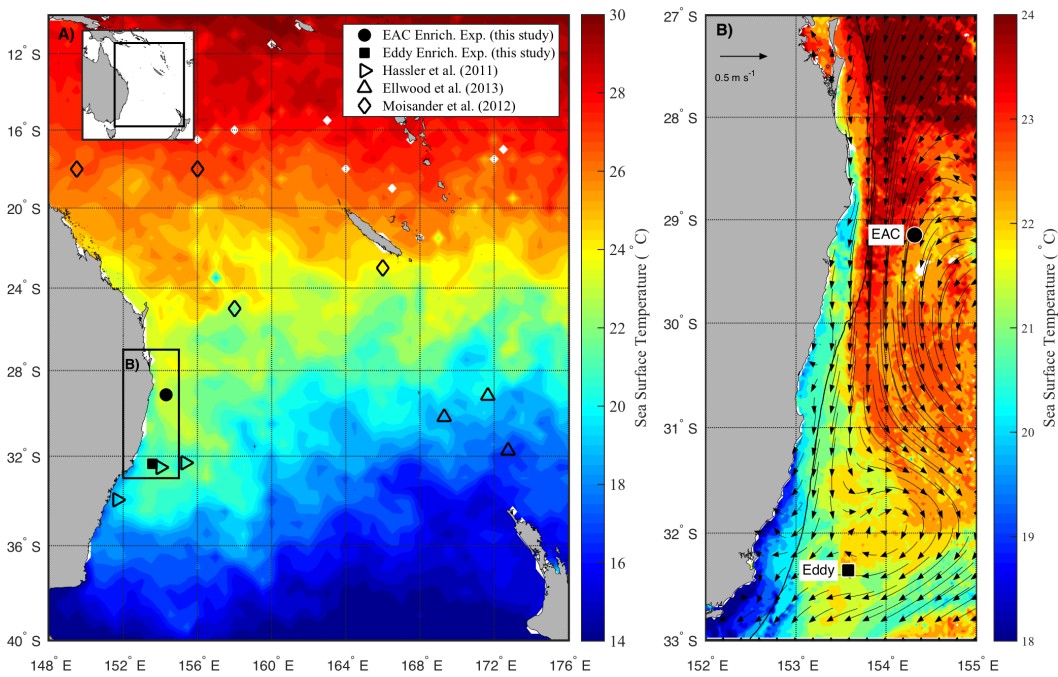

**Figure 1** **Study area.** (A) Average sea surface temperature (SST) for October 2010 in the Tasman Sea, eastern Australia. The location of previous nutrient amendment studies are shown with symbols. The black box is the domain for the current study. (B) Average SST (20–25th October 2010) of the study domain during the voyage, showing location of sampling sites. The black line shows 200 m isobath, which approximates the continental shelf edge. Geostrophic velocities, estimated from sea-level anomaly are shown as arrows (20th October 2010).

## MATERIALS AND METHODS

### Study site and water collection

The experiments were conducted on the *RV Southern Surveyor* during austral spring in 2010 (15–31 October) between 29 and 36°S, and 150 and 155°E (Fig. 1A). Sampling sites were chosen with the assistance of daily Moderate Resolution Imaging Spectroradiometer (MODIS) and Advanced Very High Resolution Radiometer (AVHRR) satellite imagery and targeted the EAC and a meso-scale cyclonic (cold core) eddy (CCE; Fig. 1B). To examine the oceanographic context of the sampling period, MODIS Level 3 sea-surface temperature was obtained from the Integrated Marine Observing System (IMOS) Data Portal (http://imos.aodn.org.au/imos/) at 1 km resolution. Satellite altimeter data were obtained from NASA/CNES (Jason-1 and 2) and ESA (ENVISAGE) via the IMOS portal.

At each site, the physico-chemical properties of the water column (0–200 m) were measured with the aid of a Conductivity-Temperature-Depth (CTD; Seabird SBE911-plus) equipped with a fluorometer (AquaTracker Mk3, Chelsea, UK), transmissometer (Wetlabs C-Star (25 cm optical path)), dissolved oxygen (Seabird SBE43) and Photosynthetically Active Radiation (PAR; Biospherical Instruments QCP-2300 Log Quantum Cosine Irradiance Sensor) sensor.

Seawater used to assess microbial composition and diversity, as well as for nutrient amendment experiments was collected from the depth of the chlorophyll-a (Chl-a)

fluorescence maximum (as determined by the down-cast fluorescence profile) using 10 L Niskin bottles (80 m in the EAC, and 41 m in the CCE). A trace metal clean rosette was not available for this voyage, so the following precautions were taken to minimise trace metal contamination. Water was sampled from Niskin bottles through acid washed silicon tubing with plastic bags covering the tubing and bottle neck, and polyethelene gloves were worn during water sampling and manipulation. Seawater was filtered through acid-soaked 200–210 µm mesh to remove mesozooplankton grazers and collected in 20 L acid washed (1 N HCl, rinsed 7-times with MilliQ water) LDPE or PC carboys. Carboys were stored in double-plastic bags and kept in plastic boxes to avoid contact with the ship. All subsequent sampling took place in a custom-made, metal free laminar flow cabinet, dedicated for trace metal clean work using standard clean room procedures.

## Experimental setup

Water was homogenised, sampled for dissolved nutrients, flow cytometry, phytoplankton pigments, nucleic acid collection and photo-physiological measurements, with the remainder transferred into pre-treated acid washed (1 N HCl, rinsed 7-times with MilliQ water) 4 L polycarbonate bottles (Nalgene) under laminar flow conditions. Bottles were randomly allocated to five nutrient addition treatments in triplicate: an unamended control, +NO$_3$ (10 µM nitrate final concentration), +NO$_3$+FeCl$_3$ (10 µM and 1 nM final concentration (ICP-MS 1g/L standard, Fluka), respectively), +Si(OH)$_4$ (10 µM final concentration), nutrient mix (+NO$_3$+Si+PO$_4$+Fe; 10N : 10Si : 0.625P µM in Redfield proportions and 1 nM Fe respectively). After the addition of nutrients, bottles were capped, gently inverted and lids sealed with parafilm, before being placed into an on-deck, flow-through incubator, exposed to 25% surface irradiance and *in situ* temperature conditions. Surface seawater supplying the deck board incubators was 21.94 ± 0.47 °C during the EAC incubation and 21.57 ± 0.27 °C during the CCE incubations. This closely matches the *in situ* temperature for both EAC (80 m) and CCE (41 m) communities at the time of sampling (Table 1).

For the EAC experiment, control bottles were sampled daily for maximum quantum yield of photosystem II ($F_V/F_M$) before being enriched with their respective nutrient or nutrient mix treatment, then re-sealed and returned to the on-deck incubator. The same sampling protocol was used in the CCE; however, no additional nutrient additions were made after the initial amendment to avoid accumulation of excess nutrients. To limit nutrient or microbiological cross-contamination, the same bottles were re-used for identical experimental (nutrient) treatments across the EAC and CCE experiments. Macronutrient analyses were carried out on board by CSIRO Marine and Atmospheric Research (CMAR) according to *Cowley et al. (1999)*. Nutrient measurements had a standard error <0.7% and a detection limit of 0.035 µM for NO$_x$, 0.012 µM for Si and 0.009 µM for PO$_4$.

After 3 days (72–78 h) of incubation, nutrient experiments were terminated. Microbial responses to nutrient additions were quantified by measuring pigments (Chl-a and other accessory pigments), abundance of pico- and nano-phytoplankton and heterotrophic bacteria via flow cytometry, and DNA sequencing for characterising prokaryote and diazotroph community diversity and composition (16S ribosomal RNA, nitrogenase *NifH* subunit targeted, respectively).

**Table 1  Starting conditions for nutrient amendment experiments in the East Australian Current (EAC) and a cyclonic cold core eddy (CCE).** Note that sampling depths targeted the chlorophyll-a fluorescence maximum (Fmax), which was deeper in the EAC than the CCE.

|  | EAC | CCE |
| --- | --- | --- |
| Location | 29.14817°S, 154.31495°E | 32.35217°S, 153.58112°E |
| Bottom depth (m) | 3,279 | 4,632 |
| Sampling depth (Fmax, m) | 80 | 41 |
| Temperature (°C) | 21.08 | 21.31 |
| Salinity | 35.52 | 35.49 |
| Ammonium ($\mu$M) | $0.07 \pm 0.02$ | $0.16 \pm 0.01$[*] |
| Nitrate ($\mu$M) | $0.26 \pm 0.24$ | $0.14 \pm 0.02$ |
| Phosphate ($\mu$M) | $0.12 \pm 0.02$ | $0.11 \pm 0.01$ |
| Silicate ($\mu$M) | $0.84 \pm 0.03$ | $0.52 \pm 0.01$ |
| Total dissolved iron (TDFe) in controls at $t72$ (nM) | $0.38 \pm 0.07$ | $1.32 \pm 0.23$ |
| Chlorophyll-a ($\mu$g L$^{-1}$) | $0.106 \pm 0.008$ | $0.336 \pm 0.041$ |

**Notes.**

[*]Analytical replicates from same CTD cast, not separate casts as for EAC ($n = 2$).

## Pigment analysis

Seawater (minimum volume 2.2 L) was filtered under low vacuum (e.g., $\leq$100 mm Hg) onto 25 mm GF/F filters in low light (<10 $\mu$mol photons m$^{-2}$ s$^{-1}$). Filters were folded in half, blotted dry on absorbent paper, placed into screw-capped cryovials and stored in liquid nitrogen until pigment analysis. In the laboratory, pigments were extracted and analysed using High Performance Liquid Chromatography (HPLC) as described in *Hassler et al. (2011)*. Biomarker pigments were used to infer the distribution of dominant phototrophs. Each biomarker pigment was normalised against Chl-a (the universal pigment in all phytoplankton) to account for spatial variation in the total phytoplankton biomass. Briefly, biomarkers represent the following phytoplankton groups: Zeaxanthin (Zea) = cyanobacteria; 19'Hexanoyloxyfucoxanthin (19-Hex) = haptophytes; Chlorophyll-b (Chl-b) and lutein = Chlorophytes; Alloxanthin (Allo) = cryptophytes, 19'Butanoloxylfucoxanthin (19-But) = pelagophytes; Prasinoxanthin (Prasino) = prasinophytes; Peridinin (Per) = autotrophic dinoflagellates; and Fucoxanthin (Fuco) = diatoms, prymnesiophytes, chrysophytes, pelagophytes and raphidophytes. The photosynthetic (PSC) and photoprotective (PPC) carotenoid pigment contributions were calculated as in *Barlow et al. (2007)*, and the approach of *Uitz et al. (2008)* was used to assess the taxonomic composition of the phytoplankton community and characterise its size structure. While this method may be subject to error because pigments are shared between different phytoplankton groups, and some groups are spread across different sizes, we apply it in this study to examine relative (not absolute) differences in size structure between water masses and nutrient treatments.

## Flow cytometry

Samples for enumeration of pico- and nano-phytoplankton were fixed with glutaraldehyde (1% v/v final concentration), snap frozen in liquid nitrogen and stored at $-80$ °C.

Populations of *Prochlorococcus*, *Synechococcus* and picoeukaryotes were discriminated using side scatter (SSC) and red and orange fluorescence (*Marie et al., 1997*) using a flow cytometer (LSR II, BD Biosciences). Pigment content per cell was normalised to the fluorescence of standard yellow-green beads (1 µm FluoSpheres®; Life Technologies) that were added to each sample immediately before analysis. Samples for bacterial analysis were stained with SYBR Green I nucleic acid stain (1:10000 final dilution; Molecular Probes) (*Marie et al., 1997*) and high and low nucleic acid content populations were discriminated according to green fluorescence and side scatter properties (*Gasol & Del Giorgio, 2000*; *Seymour, Seuront & Mitchell, 2007*). Data was analysed using Cell-Quest Pro (BD Biosciences).

## DNA extraction and sequence analysis of prokaryote and diazotrophic communities

Water samples (2 L) for DNA analyses were filtered onto 0.2 µm polycarbonate membrane filters (Millipore), snap frozen in liquid nitrogen and stored at −80 °C prior to analysis. Genomic DNA was subsequently extracted using the Power Water DNA extraction kit (MoBio Laboratories, Inc) following the manufacturer's protocols and DNA concentration was quantified using a Qubit® 2.0 fluorometer (Invitrogen). To determine bacterial community composition, the V1-V3 region of the 16S rRNA gene was amplified using the primer sets in Table S1, and sequenced by 454 pyrosequencing (Roche, FLX Titanium; Molecular Research LP) following previously published protocols (*Acosta Martinez et al., 2008*; *Dowd et al., 2008*). 16S rRNA gene sequences were analysed and processed using the Quantitative Insights into Microbial Ecology software (QIIME; *Caporaso et al., 2010*). Briefly, samples were quality filtered, de-multiplexed and clustered based on 97% sequence identity using UCLUST (*Edgar, 2010*). Taxonomy was assigned according to the latest version of the SILVA database (111; *Quast et al., 2013*) and samples were rarefied to the lowest number of sequences to ensure even sampling effort across samples (14,621 sequences per sample).

For $N_2$-fixing bacteria, the gene encoding a subunit of the enzyme nitrogenase (*NifH*, *Zehr, Mellon & Hiorns, 1997*; *Zehr et al., 2003*) was amplified and sequenced from genomic DNA using previously published methods (*Zehr & McReynolds, 1989*; *Zehr & Turner, 2001*). A nested PCR protocol was used to amplify an approximately 359 bp region of the *NifH* gene using two sets of degenerate primers listed in Table S1. PCR products were purified using the Ultra Clean PCR Clean-up Kit (MoBio Laboratories, Inc) following the manufacturer's instructions. The *nifH* amplicons were sequenced by 454 pyrosequencing (Roche, FLX Titanium; Molecular Research LP) after an additional 10-cycles PCR with custom barcoded NifH1 and NifH2 primers under the same reaction conditions (*Dowd et al., 2008*; *Farnelid et al., 2011*; *Farnelid et al., 2013*). Raw sequences were quality filtered, and de-multiplexed in QIIME (*Caporaso et al., 2010*). Sequences were clustered at 95% sequence identity using UCLUST, whereby *nNifH* sequences within 5% identity of a centroid read were assigned as operational taxonomic units (OTUs) (*Edgar, 2010*), then rarefied to 1,050 sequences per sample to ensure even sampling effort, resulting in 989 *NifH* OTUs. Since many of these *NifH* OTUs were singletons, only OTUs with ≥100 sequences

assigned to them were analysed further (representing 85% of total *NifH* sequences), resulting in 16 OTUs. Putative taxonomy was assigned using BLASTn (*Altschul et al., 1990*) against the NCBI Nucleotide collection database, and translated *NifH* sequences were compared to the Ribosomal Database Project's *NifH* protein database using FrameBot (*Wang et al., 2013*; *Fish et al., 2013*).

## Photophysiological measurements

Photosynthetic efficiency of microbial phototrophs was measured using a Pulse Amplitude Modulated (PAM) fluorometer (Water-PAM; Walz GmbH, Effeltrich, Germany). A 3 ml aliquot of water was transferred to a quartz cuvette and after a 10 min dark-adaptation period, minimum fluorescence ($F_O$) was recorded. Upon application of a saturating pulse of light (pulse duration = 0.8 s; pulse intensity > 3000 $\mu$mol photons m$^{-2}$ s$^{-1}$) maximum fluorescence ($F_M$) was determined. From these two parameters, $F_V/F_M$ was calculated according to the equation $(F_M - F_O)/F_M$ (*Schreiber, 2004*).

## Phytoplankton primary production measurements

Phytoplankton primary production was estimated at the end of the 3-day experiments using small volume $^{14}$C incubations as described in *Doblin et al. (2011)*. Carbon uptake rates were normalised to *in situ* Chl-a concentrations. Carbon fixation-irradiance relationships were then plotted and the equation of *Platt, Gallegos & Harrison (1980)* used to fit curves to data using least squares non-linear regression. Photosynthetic parameters included light-saturated photosynthetic rate [$P_{max}$, mg C (mg Chl-a)$^{-1}$ h$^{-1}$], initial slope of the light-limited section of the carbon fixation-irradiance curve [$\alpha$, mg C (mg Chl-a)$^{-1}$ h$^{-1}$ ($\mu$mol photons m$^{-2}$ s$^{-1}$)$^{-1}$], and light intensity at which carbon-uptake became maximal (calculated as $P_{max}/\alpha = E_k$, $\mu$mol photons m$^{-2}$ s$^{-1}$).

## Statistical analysis

Differences in microbial composition and diversity between the initial EAC and CCE communities, and between nutrient treatments within each water mass were assessed using analysis of variance (ANOVA; $\alpha = 0.05$). Data were analysed comparing responses at the end of incubation, $t_{72}$ (72–78 h), across treatments, as well as comparing the differences over time from the initial water ($t_0$) to $t_{72}$. Multiple comparison adjustment of the *p*-value was made using a Tukey's HSD test. To ensure that the assumption of equal variances for all parametric tests was satisfied, a Levene's test for homogeneity of variance was applied to all data *a priori* and when necessary, data was transformed. All analyses were performed using SPSS statistical software (version 22, IBM, New York USA).

To examine overall changes in microbial assemblages due to nutrient amendment, composition data (i.e., for phototrophs, diagnostic pigments standardised to total Chl-a, flow cytometric counts; for heterotrophs and diazotrophs, rarefied 16S and *NifH* sequence data, respectively) were square root transformed and a resemblance matrix was generated using Bray–Curtis similarity in the PRIMER software package (*Clarke & Warwick, 2001*). Analysis of similarities (ANOSIM) was used to test the hypothesis that different nutrient amendments would influence microbial composition (*Clarke, 1993*). The contribution of phytoplankton groups to the observed significant differences in community assemblage, as

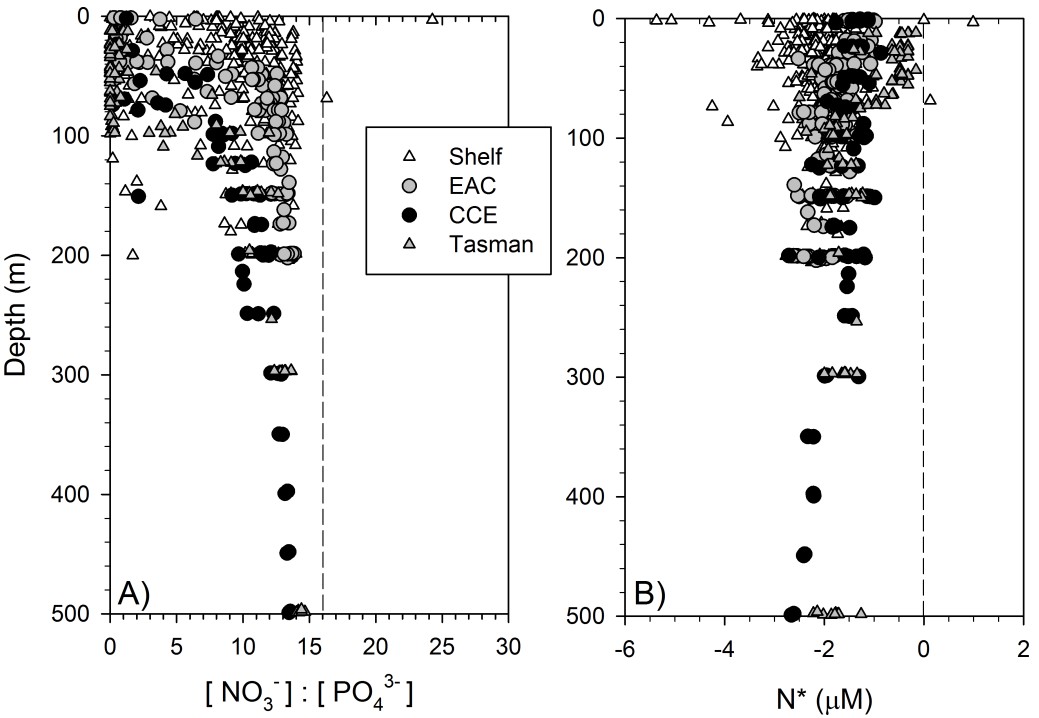

**Figure 2  Dissolved nitrogen pool relative to phosphorus.** The ratio of dissolved nitrate and phosphate (A) and nitrate deficit ($N* = [NO_3^-] - 16[PO_4^{3-}]$) (B) in waters of the study domain, including the continental shelf (white triangles), East Australian Current (EAC; grey circles), cyclonic cold-core eddy (CCE; black circles) and Tasman Sea (grey triangles).

a function of treatment, were determined using Similarity Percentage Analysis (SIMPER; *Clarke, 1993*).

## RESULTS

### Oceanographic setting

During the voyage, the EAC was flowing southward along the continental shelf edge (Fig. 1B) with a core surface temperature of 23–24 °C. The EAC surface velocity was 1.2 m s$^{-1}$ estimated from altimetry (Fig. 1B, arrows) with the current separating from the coast at ~30°S, forming the Tasman Front. The EAC station (Fig. 1B) had a temperature range of 21.4–22.5 °C and a salinity of 35.45–35.52 in the upper 200 m of the water-column. The cyclonic eddy, south of the EAC station, was sampled on 25th October 2010 when it centred at 32°S adjacent to the continental shelf. The CCE had a temperature range of 14.3–21.8 °C and a salinity of 35.26–35.54 in the upper 200 m of the water-column.

Dissolved macronutrient stocks indicated the potential for widespread N limitation in the EAC and adjacent shelf and Tasman Sea (offshore) waters (Fig. 2A), with nitrate deficit ($N* = [NO_3^-] - 16[PO_4^{3-}]$) occurring to at least 200 m (overall mean for all depths ± SD = −1.9 ± 0.53 μM; Fig. 2B). Oxidised N (nitrate) was detectable at the Chl-a fluorescence maximum in the EAC (80 m) and CCE (41 m), ranging between 0.1 and 0.3 μM (Fig. 3), with dissolved phosphate being ~0.1 μM in both water masses (Table 1).

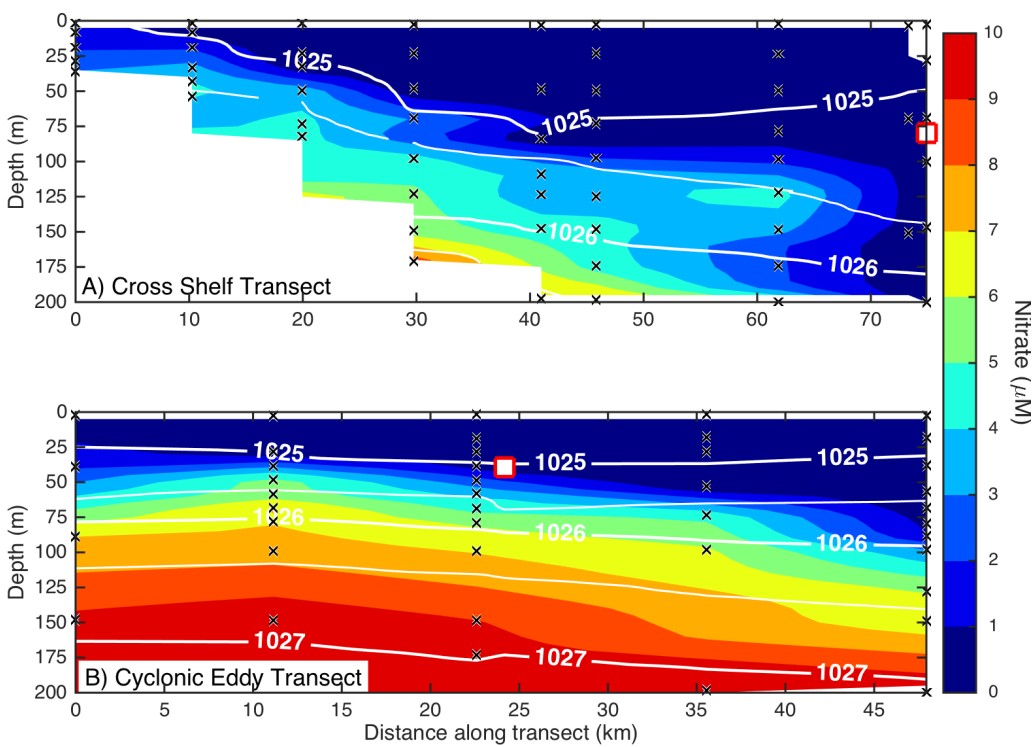

**Figure 3** **Eddy influence on vertical nutrient distribution.** The distribution of dissolved nitrate (A) across the continental shelf to the East Australian Current, and (B) across the sampled cyclonic cold-core eddy. The white contours indicate seawater density, and the black crosses show the sampling locations for nitrate. The white squares indicate the water sampling locations for the nutrient amendment experiments.

## Microbial community composition and diversity in different water masses

Initial Chl-a concentrations were relatively low (0.106 and 0.336 µg L$^{-1}$), but distinct ($p$-value <0.05), in the EAC and CCE, respectively (Table 1; t0 Figs. 4A and 4B). The most abundant phototroph in the EAC was the cyanobacterium *Prochlorococcus*, whereas in the CCE it was *Synechocccus* (Fig. S1). A significant proportion of larger phototrophs in the EAC contained 19'-hexanoyloxyfucoxanthin (Hex-Fuco; t0 Fig. 4E), exclusively found in haptophytes, including the coccolithophores (*Liu et al., 2009*). However, in the CCE, fucoxanthin (found in Phaeophyta and most other heterokonts), was the dominant accessory pigment, largely indicative of diatoms (t0 Fig. 4D). Pigment ratios suggested the size structure of the EAC phototrophic community was dominated by pico- and nano-plankton (<2 µm) and in the CCE by microplankton (≥20 µm; t0 Fig. 5A and 5B, respectively). Pigment richness was 30% higher in the CCE than in the EAC, and phototrophic alpha-diversity was ~40% higher (Shannon's index calculated using HPLC pigment data = 1.02 ± 0.05 compared to 0.64 ± 0.02 in the EAC; average ± SD here and throughout).

With respect to the heterotrophs, the total abundance of bacteria was similar in both water masses (~8.5 × 10$^5$ cells ml$^{-1}$; t0 Figs. 5I and 5J) but there was a greater proportion of high DNA bacteria in the CCE (47 ± 4 vs 39 ± 2% in the EAC). The greatest

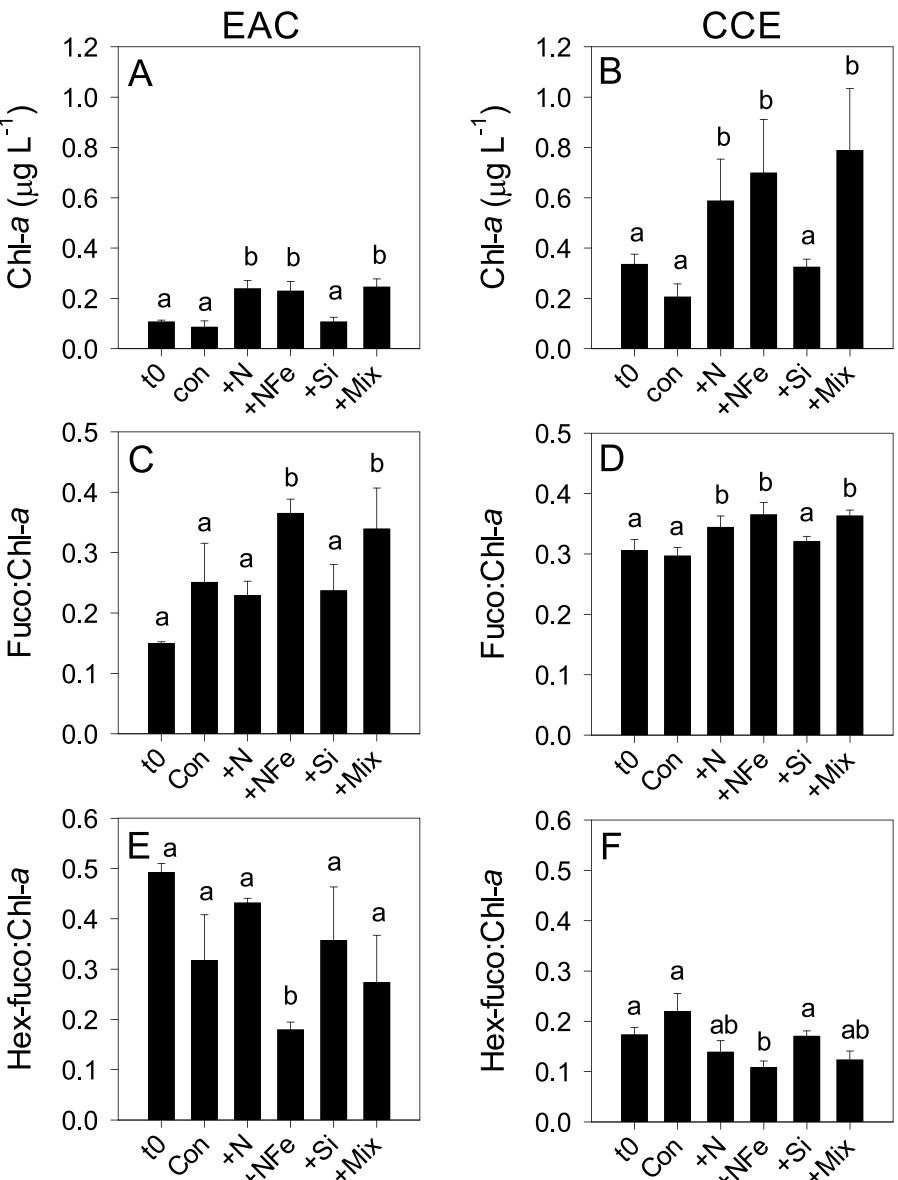

**Figure 4 Phototrophic responses to nutrient amendment.** Total Chl-a (monovinyl + divinyl), ratio of fucoxanthin to Chl-a and ratio of hex-fucoxanthin to Chl-a in the East Australian Current (EAC) (A, C and E, respecively) and cyclonic cold-core eddy (CCE) (B, D and F, respectively). These parameters are proxies for total phytoplankton biomass (Chl-a), relative biomass of diatoms (Fuco:Chl-a) and relative biomass of haptophytes (Hex-fuco:Chl-a). Treatments include initial (t0) and after 3 days nutrient amendment: Con = control, no amendment; +N = nitrate; +NFe = nitrate +iron; +Si = silicate; +Mix = nitrate + phosphate + silicate + iron. Values plotted are mean ± standard deviation. Letters above bars indicate statistical differences amongst treatments (ANOVA, $\alpha = 0.05$) such that a is different to b, and ab is the same as a and b.

proportion of bacterial 16S rRNA sequences in both water masses belonged to the alpha-proteobacteria (SAR11 and SAR116 clade), Rhodobacteriaceae, as well as *Synechococcus* and *Prochlorococcus* (Fig. 6) but there was no difference in bacterial alpha-diversity between water masses (Shannon's index in EAC = 3.20 ± 0.13 compared to 3.10 ± 0.19 in the

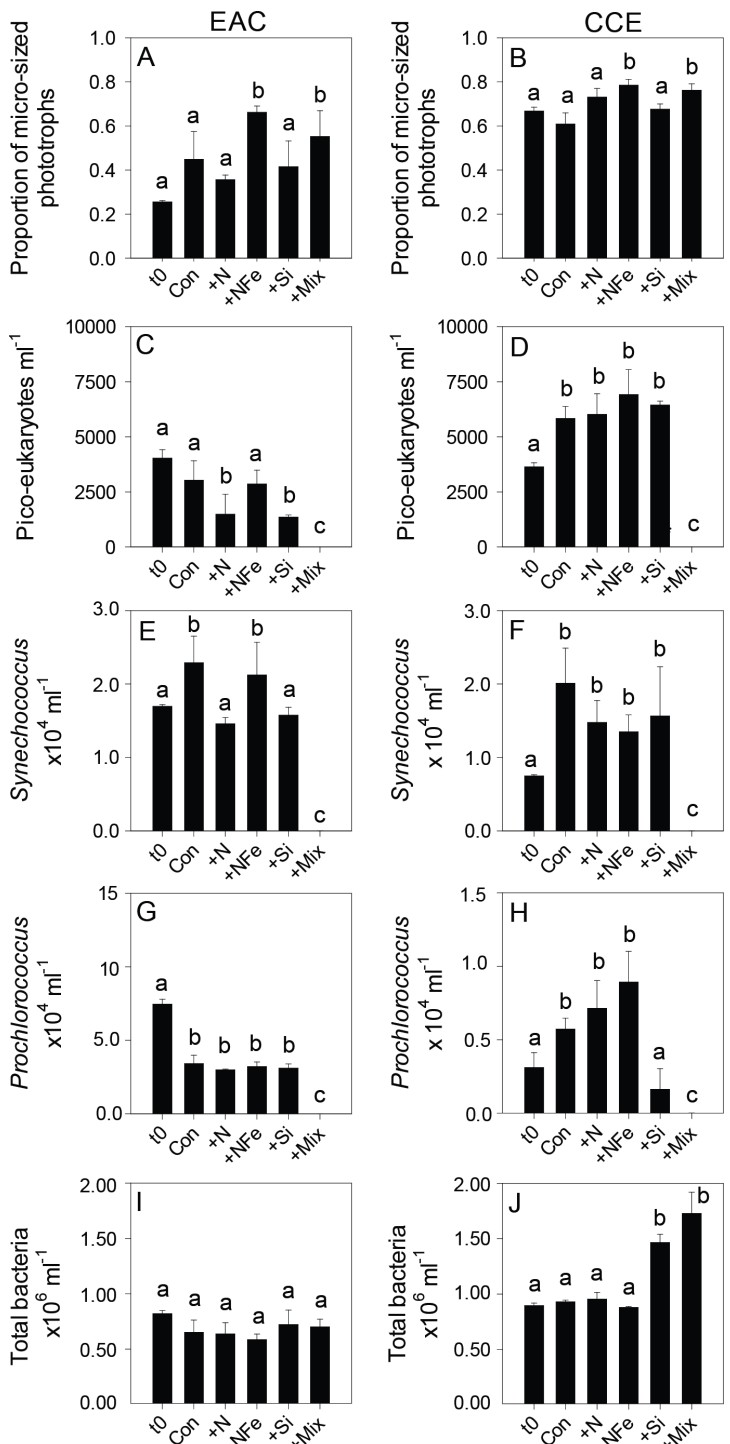

**Figure 5 Phototrophic and total bacteria responses to nutrient amendment.** Proportion of phototrophs larger than 20 μm in the EAC (A) and CCE (B), abundance of *picoeukaryotes* (C and D), abundance of *Synechococcus* (E and F), abundance of *Prochlorococcus* (G and H), and abundance of total bacteria (I and J). Treatments as in Fig. 4. Values plotted are mean ± standard deviation. Letters above bars indicate statistical differences amongst treatments (ANOVA, $\alpha = 0.05$) such that a is different to b and c.

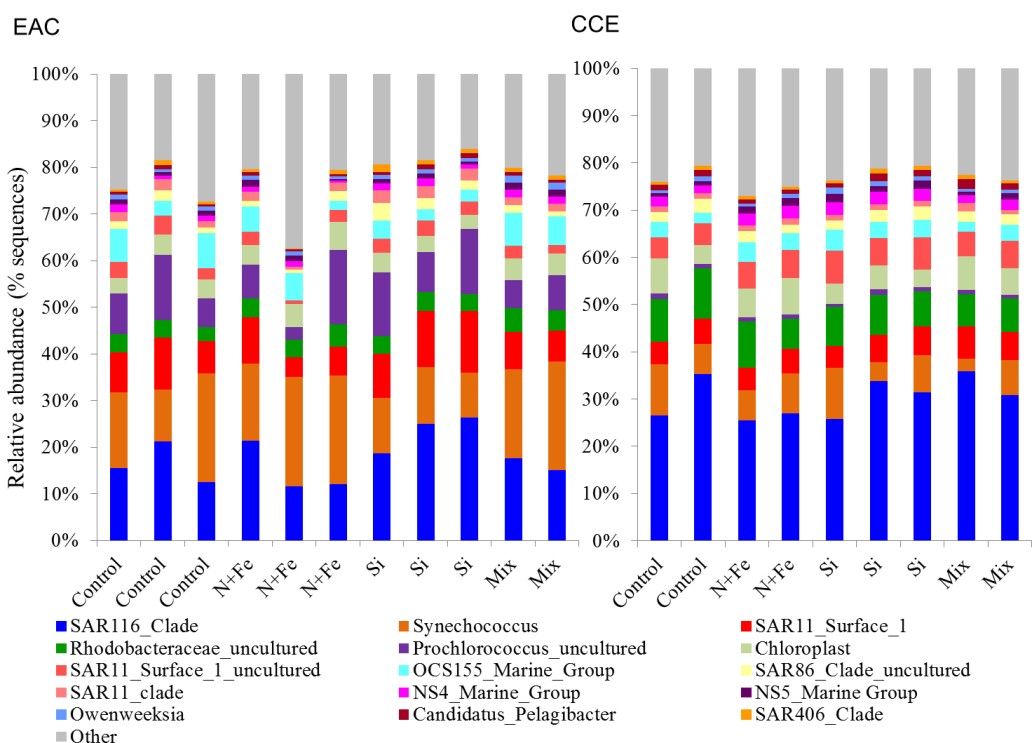

**Figure 6 Relative abundance of 16S rRNA operational taxonomic units.** Data are shown for sequences with ≥ 97% sequence similarity to the SILVA database in the EAC (A) and CCE (B) amongst different treatments. For visual simplicity, only the top 15 OTUs are presented, with the upper grey bars representing the remaining 16S sequences detected. Control = no amendment; N + Fe = addition of nitrate and iron (10 µM and 1 nM, respectively), Si = addition of silicate (10 µM) and Mix = addition of nitrate, phosphate, silicate and iron (10 µM, 0.625 µM, 10 µM, 1 nM, respectively).

CCE). Heterotrophic *NifH* gene sequences were detected in the EAC and were primarily attributable to members of the gamma-, beta-, and delta-proteobacteria. Initial CCE samples were not available (cryovials broken in storage); however control CCE treatments contained phototrophic and heterotrophic diazotrophs, including *Trichodesmium*, UCYN-A and gamma-proteobacterial *NifH* sequences.

## Nutrient-induced shifts in the phototrophic community

After 3 days, there was a large positive effect of NFe and the nutrient Mix (N, P, Si, Fe) on the total Chl-a concentration in the EAC community (Fig. 4A), which occurred alongside an increase in fucoxanthin relative to Chl a in the NFe and Mix treatments (*p*-value ≤ 0.026; Fig. 4C), suggesting a greater relative abundance of diatoms (Table 2). The relative abundance of haptophytes (as indicated by pigment Hex-Fuco) showed a declining trend across all treatments, but by the end of the experiment haptophytes were least abundant in the NFe treatment (*p*-value ≤ 0.031; Fig. 4E). *Prochlorococcus* (as determined by flow cytometric counts) decreased in all treatments including the controls (Fig. 5G), but *Synechococcus* abundance declined only in the Mix treatment (Fig. 5E). Collectively, there was a decrease in picoeukaryote abundance in all but the NFe EAC bottles by day 3 (*p*-value < 0.05; Table 2), with the phototrophic community overall showing a significant

**Table 2  Effect of experimental manipulation on microbial assemblages in the East Australian Current (EAC) and a cyclonic cold core eddy (CCE) as shown by comparison of $t_0$ with $t_{72}$ no amendment control, as well as $t_{72}$ nutrient amendments relative to controls.** Treatments include NO$_3$ (10 μM nitrate final concentration), NO$_3$+ Fe (10 μM nitrate and 1 nM Fe final concentration), Si (10 μM final concentration), and Mix (N + Si + P + Fe; 10N : 10Si : 0.625P μM in Redfield proportions and 1 nM Fe respectively). ++ strong positive difference, $p < 0.01$; + positive difference, $P < 0.05$; − strong negative difference, $P < 0.01$; −negative difference, $P < 0.05$; blank cells: no significant difference; nd: not detected; shaded cells: no measurement.

| Parameter | $t_0$ vs ctrl EAC | $t_0$ vs ctrl CCE | NO$_3$ EAC | NO$_3$ CCE | NO$_3$+Fe EAC | NO$_3$+Fe CCE | Si EAC | Si CCE | Mix EAC | Mix CCE |
|---|---|---|---|---|---|---|---|---|---|---|
| Chlorophyll a | | - | + | + | + | + | | | + | + |
| Fucoxanthin:Chl a | | | | + | + | + | | | + | + |
| Hex-Fuco:Chl a | - | | | | - | - | | | | |
| Peridinin:chl a | nd | | nd | - | nd | - | nd | - | | |
| Cell abundance* | | | | | | | | | | |
|   – Total pico and nano eukaryote | − | + | | | | | | | − | |
|   – *Prochlorococcus* | − | | | | | | | | − | |
|   – *Synechococcus* | + | + | | | | | | | − | − |
|   – Picoeukaryote | | | − | | | | − | − | | |
|   – Nanoeukaryote | | | | | | + | | | | |
|   – Bacteria | | | | | | | | ++ | ++ | |
| $F_v/F_M$ | | | - | | | | | | | |
| Chl-a fluorescence/cell | − | | ++ | ++ | ++ | − | − | | | − |
| Phycoerythrin fluorescence/cell | | | ++ | | ++ | | | | | |
| Primary production | + | | ++ | ++ | − | ++ | | | | |
| alpha | | | | ++ | | ++ | | | | |
| Ik | | | | − | | ++ | | | | |
| Growth rate | | | | | | | | | | |
|   – Total | | | | | | | | | − | |
|   – *Prochlorococcus* | | | | | | | − | | | − |
|   – *Synechococcus* | | | − | | | | | | − | |
|   – Picoeukaryote | | | − | | | | − | | − | |
|   – Nanoeukaryote | | | | | | + | | | | |
|   – Bacteria | | | | | | | | ++ | | ++ |

**Notes.**
*Estimated from flow cytometry.

cell size increase into the micro size class (nominally ≥ 20 μm) with N amendment ($p$-value < 0.05; Fig. 5A), as estimated through pigment ratios (*Uitz et al., 2008*). Taken as a whole, the largest shift in phototrophic community composition and structure was in the mix treatment relative to all other treatments (SIMPER, >75% dissimilarity), which was attributed mainly to the decrease in *picoeukaryotes*. Importantly, the EAC community was more similar to the $t_0$ CCE community after nutrient amendment, particularly with NFe addition (Global R = 0.48, $p$-value < 0.05; Fig. 7C).

In contrast, within the CCE, addition of N, NFe and Mix caused a significant increase in Chl-a relative to the initial community (Fig. 4B; Table 2). Consistent with the patterns seen

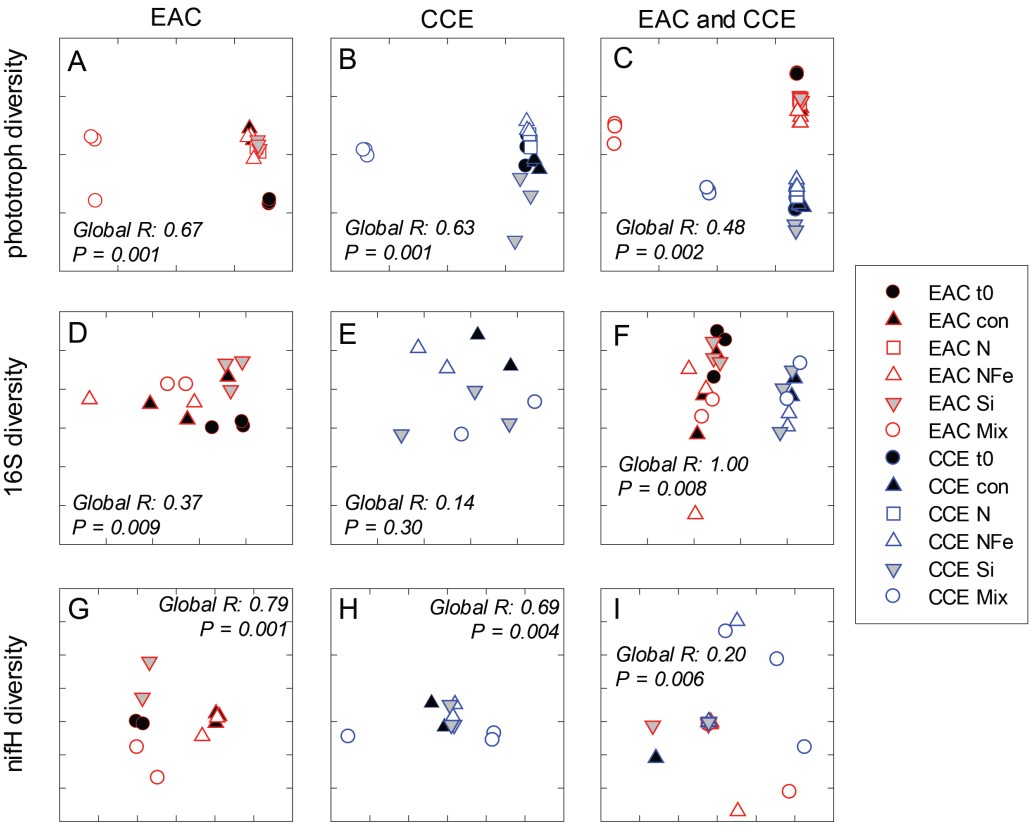

**Figure 7 Diversity of microbial communities.** Multi-dimensional Scaling (MDS) plots of phototrophs (A, B and C), bacteria (D, E and F) and diazotrophs (G, H and I) in the East Australian Current (EAC), cyclonic cold-core eddy (CCE) and both the EAC and CCE. Clustering of samples is based on a Bray-Curtis similarity matrix of square-root transformed HPLC pigment concentrations and flow cytometric counts of phototroph abundance, operational taxonomic units from 16S ribosomal genes or nitrogenase *NifH* subunit genes. Plots on the same row have the same axes scales, to make them directly comparable. Stress values for all plots are <0.10.

in the EAC, Fucoxanthin:Chl-a increased in NFe and Mix bottles (and in the N treatment), which was concomitant with a decline in haptophytes (Hexo-Fuco:Chl-a; Figs. 4D and 4F; Table 2). There was a decrease in *picoeukaryotes* in the Mix treatment relative to unamended controls (Fig. 5D), as well as a measurable increase in *Prochlorococcus* with NFe amendment (Fig. 5H). *Synechococcus* showed little effect of nutrient amendment in the CCE but declined when all nutrients were added together ($p$-value < 0.05; Fig. 5F; Table 2). Similar to the EAC, there was an overall increase in cell size of the CCE phototrophic assemblage into the micro size class with NFe and Mix amendment ($p$-value < 0.05; Fig. 5B). The Mix addition caused the greatest shift in cell size structure, decreasing the abundance of small cells by an order of magnitude (Fig. 5). Similar to the EAC community, the largest shift in phototrophic community composition and structure was in the Mix relative to all other treatments (SIMPER, ≥55% dissimilarity), mainly due to the decrease in *Synechococcus* and *picoeukaryotes*.

## Nutrient-induced shifts in the heterotrophic community

Nutrient-induced changes in the EAC bacterioplankton community were also detected by the end of the experiment (Figs. 5I, 6, 7). There was a significant (but relatively small) decrease in the relative abundance of sequences matching SAR11 Surface 1 and SAR116 clades, and an increase in relative abundance of *Synechococcus* with N, NFe, Mix addition, with the greatest dissimilarity in bacterial community composition observed between the $t_0$ and NFe treatments (SIMPER, 29% dissimilarity). These shifts in SAR11, SAR116 and *Synechococcus* relative abundance contributed to 3, 2 and 2% dissimilarity between the $t_0$ and NFe treatments respectively (Fig. 6; SIMPER). This pattern of increasing *Synechococcus* sequence abundance was consistent with the patterns of *Synechococcus* abundance revealed from flow cytometry (Fig. 5E). Within EAC nutrient addition treatments, the greatest dissimilarity was observed between bacterioplankton communities in the NFe and Si amendments (SIMPER, 29% dissimilarity). A relative decrease in *Synechococcus* with Si, and an increase in the SAR116 clade and *Prochlorococcus*, contributed 3, 2 and 2% to the dissimilarity between NFe and Si treatments, respectively (Fig. 6; SIMPER).

In the CCE bacterioplankton composition was distinct from the EAC in unamended controls after 3 days (ANOSIM, Global R: 1.00, *p*-value < 0.01; Fig. 7F), with the SAR116 clade and Rhodobacteraceae more abundant in the CCE than in the EAC, and SAR11 surface clade and *Prochlorococcus* less abundant (Fig. 6A). Total heterotrophic bacterial abundance in the CCE doubled after 3 days with Si and Mix addition, unlike the EAC community (Figs. 5I and 5J). However, there was no appreciable shift in bacterioplankton diversity between nutrient addition treatments (Fig. 7E; ANOSIM, Global R: 0.14, *p*-value > 0.05).

Similar to the patterns observed in the overall bacterioplankton compositon, diazotroph diversity in the EAC shifted following nutrient amendment (ANOSIM, Global R: 0.79, *p*-value = 0.001; Fig. 7G). Nutrient addition shifted the composition of diazotrophs from gamma-, beta-, and delta-proteobacteria toward several Cluster 1 gamma-proteobacterial *NifH* sequences (*NifH* OTU 608, 2012 and 95), sharing ≥ 89% amino acid identity with *Pseudomonas stutzeri* (*Moisander et al., 2014*; *Moisander et al., 2012*), which comprised negligible proportions of initial EAC *NifH* sequences. Shifts in the relative abundance of the three different gamma-proteobacterial *NifH* OTUs contributed to a substantial proportion of the dissimilarity between diazotroph communities detected in EAC nutrient treatments. For example, *NifH* OTU608 dominated NFe bottles, comprising up to 97% of *NifH* sequences and was responsible for 33% of the average dissimilarity between $t_0$ and NFe (SIMPER, 83% total dissimilarity). In Si bottles, *NifH* OTU2012 represented up to 87% of *NifH* sequences detected, and contributed 41% to the average dissimilarity between the $t_0$ and Si bottles (SIMPER, 65% dissimilarity). Similarly *NifH* OTU95 comprised up to 89% of *NifH* sequences in the Mix treatment and was responsible for 22% of the dissimilarity between the $t_0$ and diazotrophs in the Mix addition.

In contrast to *NifH* sequences retrieved from EAC nutrient treatments, the dominant *NifH* OTU across CCE nutrient amendments was OTU2331, which shared 90% amino acid identity to the genus *Coraliomargarita* of the Verrucomicrobia. In addition, sequences sharing 95% *NifH* amino acid identity with *Trichodemsium erythraeum* (OTU181) and *Candidatus Atelocyanobacterium thalassa* (UCYN-A; OTU1321 and OTU50), were detected

in the unamended CCE diazotroph assemblages, but no such cyanobacterial *NifH* sequences were present in the EAC. Differences in the relative abundance of these cyanobacterial diazotrophs were observed between treatments, such as an increase in OTU181 from a maximum of 15% in the control to 62% in the Si treatment, there were also significant shifts in diazotroph composition between nutrient addition treatments (Fig. 7H) (ANOSIM, Global R: 0.69, *p*-value = 0.004). Notably, there were no *NifH* sequences detected in the CCE Mix treatment.

## Time-dependent response to nutrient amendment: subcellular to community-level

Daily sampling of incubation bottles following nutrient amendment captured a cascade of effects at the cellular, population and community level, indicating differences in the speed of response by different microbes to nutrient resupply. Despite lower phytoplankton biomass, the EAC community took up ~10 times more nitrate than the CCE community within the first 24 h (Figs. 8E and 8F). Net nitrate uptake in the EAC (averaged over day 1–3) was greatest in the Mix ($2.98 \pm 0.88 \; \mu M \; d^{-1}$), and NFe and N treatments (~2 $\mu M$ $d^{-1}$), but was <0.3 $\mu M \; d^{-1}$ in all other treatments. In the CCE, the level of nitrate uptake was an order of magnitude lower, with rates ranging from 0.02 to 0.25 $\mu M \; d^{-1}$ (in the control and Mix, respectively). Net phosphate uptake by both communities was relatively constant across nutrient amendments (~0.1 and ~0.01 $\mu M \; d^{-1}$ in the EAC and CCE, respectively), with the exception of 30% higher rates in the EAC Mix treatment (Fig. S3). Net silicate uptake rates in the EAC were greater than the CCE (~0.1 vs ~0.05 $\mu M \; d^{-1}$, respectively), with rates increasing significantly ($\geq 6$ times) when Si was added alone or with N, P and Fe (i.e., Mix treatment; Fig. S3). Total dissolved iron (TDFe) concentrations at the end of the experiment were $0.38 \pm 0.07$ nM in the unamended EAC control bottles, and $1.32 \pm 0.23$ nM in the CCE controls (Fig. S3), suggesting a strong Fe consumption and thus a limited Fe potential contamination in the experiment (spikes were 10 nM).

To illustrate the impact of nutrient amendment at the cellular, population and community level in the EAC and CCE, Fig. 8 shows time-dependent responses to the N treatment. The cellular pigment content (as estimated using flow cytometer fluorescence emission detected at red/orange wavelengths, corresponding to Chl-a/phycoerythrin) of EAC phototrophs in the N treatment was relatively constant over the first 48 h, but then increased in both phycoerythrin-containing prokaryotes and Chl-a-containing cells by $t_{72}$ (Fig. 8A; Tables 1 and 2). In the CCE, N addition caused rapid synthesis of Chl-a (but not phycoerythrin) in the first 24 h. By day 3 however, phycoerythrin-containing prokaryotes in the CCE had doubled their pigment content, in a manner that was similar to the observations in the EAC (Fig. 8B).

The initial ranked order abundance of microbial populations (as determined flow cytometrically) was different in the EAC and CCE and showed variable temporal dynamics. N amendment of the EAC community resulted in a doubling of total bacteria abundance within the first 24 h (Fig. 8C), but they then decreased to initial concentrations by $t_{72}$. The abundance of all but the large picoeukaryote population declined in the EAC under N amendment, resulting in overall negative phototrophic growth over the three day

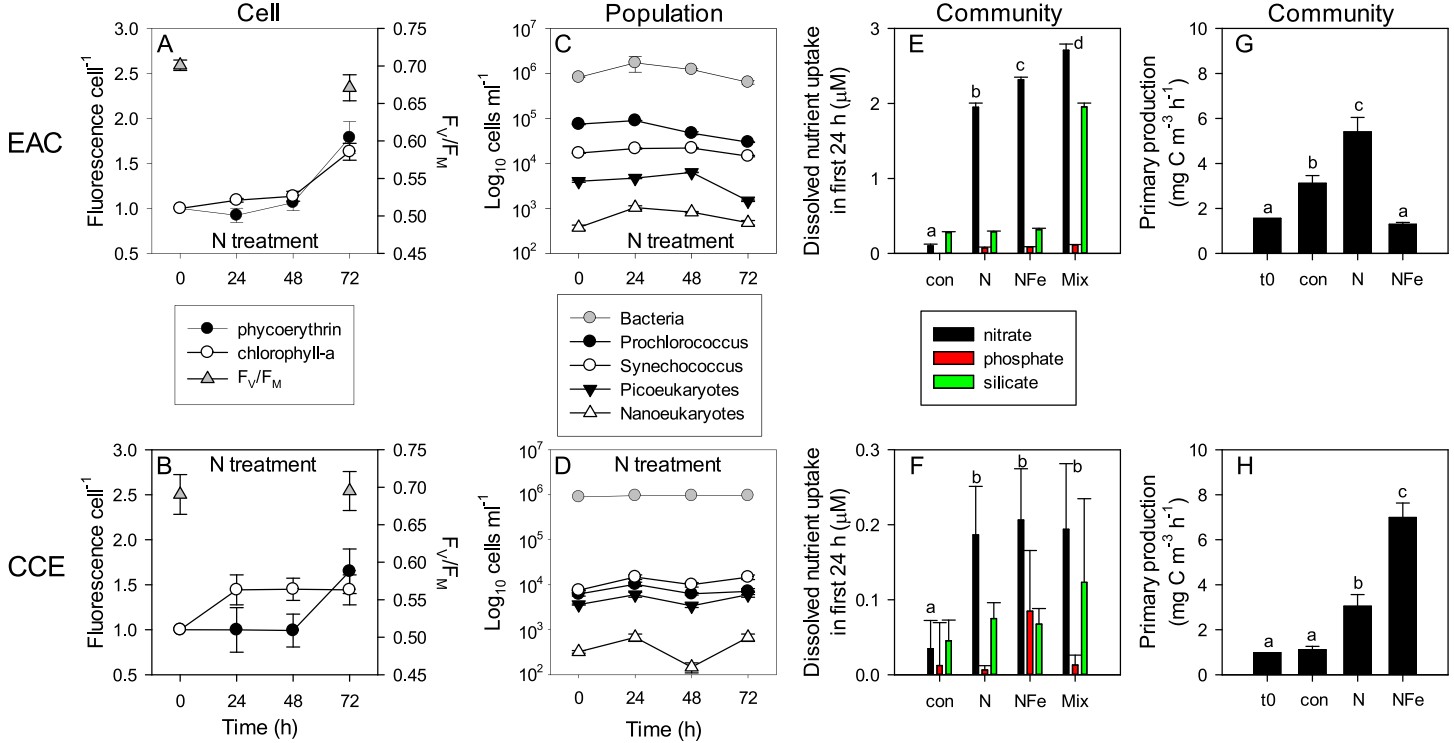

**Figure 8** **Time-course of microbial responses to nutrient addition.** Daily chlorophyll-a and phycoerythrin fluorescence in small picoeukaryotes and *Synechococcus*, respectively under N amendment (A and B), together with photosynthetic efficiency ($F_V/F_M$) of the control phytoplankton assemblage at $t_0$ and $t_{72}$; daily abundance of phototrophic and bacterial populations (C and D) in +N treatments in the EAC and CCE; rate of macronutrient uptake in the first 24 h of incubation (E and F); total carbon fixation by the phytoplankton assemblage in different treatments on day 3 (G and H). Values plotted are mean ± SD ($n = 3$) except for plots C and D which are mean ± SE ($n = 3$). Letters above bars indicate statistical differences amongst treatments (ANOVA, $\alpha = 0.05$) such that a is different to b, c and d.

experiment ($-0.153 \pm 0.016$ $d^{-1}$; Table S2). In contrast, the abundance of all CCE phototroph populations increased within the first 24 h following N amendment (Fig. 8D), and yielded significant positive growth during the experiment ($0.262 \pm 0.043$ $d^{-1}$).

Nutrient amendment caused a minor decline in the maximum quantum yield of PSII ($F_V/F_M$) in the EAC microbial community with N addition ($p$-value $= 0.013$; Table 2), but had no effect on $F_V/F_M$ in the CCE community, remaining >0.65 in the unamended controls (Figs. 8A and 8B). The light saturating irradiance $I_K$ was ~50% lower in the EAC, reflecting the greater depth of the sampled community (Table 1), and despite lower phytoplankton biomass, carbon fixation rates in unamended EAC bottles were higher than that of CCE controls (Fig. 8G and 8H). N amendment of the EAC community lead to a significant increase in light utilisation efficiency ($\alpha$) and light saturated photosynthetic rate ($P_{max}$) with no change in minimum saturating irradiance ($I_K$) between the control and N enriched cells. There was however, a significant decline in $I_K$ in the NFe treatment ($p$-value $< 0.05$; Table 2). In the CCE, there was a significant increase in $\alpha$ and $P_{max}$ with both N and NFe addition, with $I_K$ lowest in the NFe treatment. There was a 1.7 times increase in the maximum rate of primary productivity in the N amended treatment for the EAC ($p$-value $< 0.001$), compared with a 2.7 times increase in the CCE ($p$-values $< 0.001$; Fig. 8G and 8H).

Co-addition of NFe resulted in a decrease in primary productivity in the EAC (0.4 fold change), but a 6.28 fold increase in primary productivity in the CCE. This suggests stronger limitation of primary productivity in the CCE, with a greater proportional increase in $P_{max}$ with NFe compared to N addition.

## DISCUSSION

The intensification of western boundary currents in the global ocean (*Wu et al., 2012*) raises significant questions about their impact on microbial composition and biogeochemical activity, particularly in light of the potential for WBCs to promote meso-scale eddy formation (*Mata et al., 2007*) and induce nutrient upwelling (*Roughan & Middleton, 2002*). Here we show clear differences in the vertical nutrient structure of a cyclonic cold-core eddy relative to adjacent waters, and an increase in microbial diversity and size structure of an eddy assemblage relative to an adjacent western boundary current, the EAC. Our results indicate that cyclonic eddies increase delivery of N to the upper ocean but also increase the biological demand for Fe that is necessary to sustain the growth of large-celled phototrophs and potentially support the diversity of diazotrophs over longer time-scales.

### Responses of microbes to nutrient amendment

Previous studies in the Tasman Sea region have determined that surface phytoplankton communities are limited by N (*Hassler et al., 2011*; *Hassler et al., 2014*; *Ellwood et al., 2013*), and that diazotrophs (bacteria that are able to bypass nitrate limitation by fixing atmospheric N) inhabit waters to ∼100 m following nutrient draw-down over the summer (*Moisander et al., 2010*). Our observations show that EAC phytoplankton sampled at 80 m below the surface are also N limited, and that upwards displacement of isopycnal surfaces induced by a cyclonic eddy introduces nutrients into the euphotic zone to alleviate potential N limitation (Fig. 4). Phytoplankton in the CCE (sampled at ∼40 m) were also limited by N, but rates of carbon fixation more than doubled under NFe relative to N amendment, likely due to the different species composition underpinning primary productivity. In both water masses, a positive response in Chl-a was also seen with addition of N + Fe + Si + P, suggesting co-limitation by multiple nutrients in both water masses, likely a result of different nutrient requirements by different phototrophs. Uplift of N together with Fe and other nutrients would therefore increase the rate of biomass production in the eddy as well as increase the relative abundance of diatoms, both of which could act to increase export production relative to adjacent waters.

While the starting communities in the EAC and CCE were distinct, microbial phototrophs showed similar responses to nutrient amendment in both water masses, with NFe addition to the EAC resulting in phototrophs more similar to those in the initial CCE community (Fig. 7C). Fucoxanthin containing cells (likely diatoms that contribute significantly to export production; *Honjo et al., 1995*) became more prevalent in NFe and Mix treatments, whereas haptophytes clearly diminished under these amendments. Similarly, peridinin-containing dinoflagellates were initially present in both communities, but their abundance declined with nutrient amendment, becoming undetectable amongst other EAC microbes at the end of our experiment. *Prochlorococcus* was an order of

magnitude more abundant in the EAC relative to the CCE, consistent with the warmer "tropical" signature of this western boundary current (*Seymour et al., 2012*), and declined in abundance in the Mix treatment. In the CCE samples, *Prochlorococcus* similarly decreased in the Si and Mix treatments. *Synechococcus* had a positive response to bottle enclosure but unlike other studies (*Moisander et al., 2012*), we found that its abundance was similar in control and enriched treatments. Maximum net growth rates were observed in the eddy NFe bottles for *Prochlorococcus* ($0.345 \pm 0.047 \ d^{-1}$), small *picoeukaryotes* ($0.212 \pm 0.033 \ d^{-1}$) and large *picoeukaryotes* (i.e., nano-eukaryotes; $0.297 \pm 0.059 \ d^{-1}$), but for *Synechococcus*, maximum growth was found in CCE controls ($0.324 \pm 0.047 \ d^{-1}$, relative to $0.194 \pm 0.035 \ d^{-1}$ with NFe amendment).

Among the heterotrophic bacteria, initial EAC and CCE populations displayed similar abundance and alpha-diversity, and nutrient amendment lead to compositional shifts in both water masses. Addition of nitrate to the EAC community caused a doubling of total bacterial abundance within the first 24 h. Bacteria also became more abundant in the CCE community after 3 days within the Si and Mix bottles. Elevated bacterial abundance is likely to have arisen through increased dissolved organic matter (DOM) production by the resident community via several potential mechanisms: (1) mortality and cell lysis; (2) elevated rates of DOM release due to nutrient supplementation; and (3) altered microbial composition. While we did not measure rates of DOM production, our data certainly demonstrate coupling between autotrophs and heterotrophs, just as in a previous study (*Baltar et al., 2010*).

## Diazotroph relative abundance and diversity

Across the global ocean, the main supply of nitrogen into surface waters is via transport from below the thermocline, but in many regions significant amounts of new nitrogen may also be supplied via nitrogen fixation (*Capone et al., 2005*; *La Roche & Breitbarth, 2005*). Our molecular analyses revealed the presence of heterotrophic gamma-proteobacterial diazotrophs in the sub-surface EAC microbial community, whereas cyanobacterial diazotrophs (i.e., *Trichodesmium* and UCYN-A) were detected in the CCE. Given the relatively high iron requirement of diazotrophs (*Kustka et al., 2003*), as well as their low reliance on dissolved inorganic N, we expected to see a divergent response of the diazotrophs to nutrient amendment. Indeed, in the Mix bottles, diazotrophs became undetectable in the CCE samples, and were clearly outcompeted by the eukaryote phototrophs. Nutrient addition resulted in *NifH* OTU2331 becoming more prominent across CCE nutrient amendments. This OTU shares 90% amino acid identity to the genus *Coraliomargarita* of the Verrucomicrobia, and was not present amongst the EAC diazotrophs. In the EAC, nutrient addition caused a shift from the mixed proteobacterial community observed at t0, to *NifH* OTU608, *NifH* OTU2012 and *NifH* OTU95. These OTUs represent distinct taxa at the 95% sequence similarity level, yet share the same percent identity at the amino acid level, falling within the same clade of Cluster 1 gamma-proteobacterial diazotrophs. The physiology or genome content of these taxa remains completely unknown, but they seem to be abundant in warm, oligotrophic surface waters globally (*Langlois, LaRoche & Raab, 2005*). In the southwestern Pacific, the abundance of this group has been shown to

be correlated to DOC concentration, and to increase with Fe and P addition (*Moisander et al., 2012*). In contrast, the dominant diazotroph in the CCE was an OTU sharing 95% *NifH* amino acid identity with *Trichodemsium erythraeum* (OTU181) and OTUs matching *Candidatus Atelocyanobacterium thalassa* (UCYN-A; OTU1321 and OTU50). UCYN-A is the dominant diazotroph in the Coral Sea (source of the EAC) during the austral spring (*Messer et al., 2015*), and is also relatively abundant in the western South Pacific (*Moisander et al., 2010*). UCYN-A abundance has previously been shown to increase in response to Fe and organic carbon additions (*Moisander et al., 2012*), while Fe has also been shown to increase the rate of $N_2$ fixation by UCYN-A (*Krupke et al., 2015*).

Our observations demonstrate the presence of heterotrophic $N_2$-fixing organisms in sub-surface waters with strong nitrate deficiency (but not necessarily low nitrate concentration; Fig. 2); however, a remaining research need is to conduct $N_2$-fixation rate measurements in conjunction with diazotroph diversity assessment to verify whether this process represents a significant source of new N within and outside eddies, and whether changes in $N_2$-fixation are due to a shift in diversity or a change in *NifH* expression.

## Physiological responses to nutrient supply

At the cellular level, N addition caused rapid pigment synthesis (doubling of normalised fluorescence per cell) in the CCE assemblage within the first 24 h after amendment, indicating that nutrient uplift could initially cause 'greening' in the absence of an increase in cellular biomass (*Behrenfeld et al., 2015*). By the end of our experiments, both phycoerythrin and Chl-a quotas increased in both water masses with N amendment, concomitant with an increase in total Chl-a, suggesting both pigment synthesis and biomass production contributed to elevated Chl-a. These observations are of obvious importance for the accurate interpretation of satellite and other *in situ* Chl-a fluorescence data within meso-scale eddy features.

Photophysiological changes such as a decline in PSII turnover time, cross-sectional area and increased electron transport rates have been detected upon relief of nutrient limitation (*Milligan, Aparicio & Behrenfeld, 2012*). We detected minimal change in the photochemical efficiency (maximum quantum yield of PSII) with nutrient amendment, but there were measureable changes in the shape of the photosynthesis-irradiance (P–I) curve. Nitrate amendment of the EAC community increased light utilisation efficiency and light saturated photosynthetic rates, indicating increased photosynthetic electron transport and antenna size. Our flow cytometry data suggest that this was likely due to increased chlorophyll quota rather than an increase in cross sectional area (Fig. 7D). In the CCE community, the greatest change in carbon fixation (P–I) parameters was measured in the NFe treatment, suggesting an increase in PSII units per cell (detected via increased pigment quotas after 24 h), as well as an increase in total photosynthetic biomass, resulting in increased electron transport rates and light capturing capacity.

## Nutrient uptake dynamics

Initial uptake rates of nitrate (i.e., within the first day) by the EAC community were 10 times higher than in the CCE despite 3-fold lower phytoplankton biomass. This is consistent

with the opportunistic nutrient uptake strategies of phytoplankton in oligotrophic habitats (*McCarthy & Goldman, 1979*) and the theory that fast growing algae (with small cell size) are stimulated by short-term nutrient supply (*Pedersen & Borum 1996*). However, it is unlikely that such high nutrient uptake rates would be sustained in the EAC, largely because nutrient inputs (of the magnitude used in our experiments) are episodic and often coincide with major physical disturbances such as cyclones (e.g., *Law et al., 2011*). A comparison of the initial EAC and CCE communities suggests that prolonged nutrient inputs within cyclonic eddies results in a shift toward larger cells which generally have greater capacity for nutrient storage and higher nutrient requirements for growth (*Litchman et al., 2007*).

Despite these potential experimental artefacts, the relative differences in nutrient demand between treatments show that maximum net uptake rates in both water masses occurred when all macronutrients were added together, compared to treatments which contained a surplus of N, P or Si relative to other macronutrients and iron. Under N amendment, uptake ratios of N:P in the EAC were ~20 compared to ~10 in the CCE, and uptake ratios of N:Si were ~10 and 4 in the EAC and CCE, respectively, but they were closer to Redfield (Si:N:Pi = 1:1:16) in the Mix treatment (Fig. S2). This indicates that the vertical distribution of nutrients relative to one another will regulate microbial responses to eddy-induced uplift, as has been shown by *Bibby & Moore (2011)* with respect to N:Si in the sub-tropical north Atlantic and central Pacific near Hawai'i.

## CONCLUSIONS

Phytoplankton community structure plays an important role in the ecology and biogeochemistry of pelagic ecosystems including the export of organic matter to the deep ocean and the sequestration of carbon (*Follows & Dutkiewicz, 2011*; *Karl et al., 2012*). Here we show that cyclonic eddies enhance primary production in this WBC region by delivering nitrate to the upper ocean. The enhanced productivity was driven largely by an increase in the abundance of diatoms, with a concomitant decline in the abundance of haptophytes and peridinin-containing dinoflagellates.

This study confirms the low-nutrient low Chl-a status of the Eastern Australian Current (EAC) to sub-surface depths of ~80 m, and provides the first evidence of N and Fe co-limitation in an adjacent cyclonic eddy, demonstrating that such meso-scale features have the potential to increase internal nutrient inputs into the upper ocean and thereby change microbial composition and nutrient demand. Importantly, eddies may provide a critical compensatory mechanism to enrich the upper ocean and counteract increasing stratification occurring under climate change (*Matear et al., 2013*). The divergent response of large phototrophs and diazotrophs in our nutrient amendment experiments suggests that $N_2$-fixing cyanobacteria and heterotrophic bacteria are an important functional group to include in biogeochemical models, whose abundance and diversity have been under appreciated in this region until recently (*Messer et al., 2015*).

## ACKNOWLEDGEMENTS

We thank the officers and crew of the *R/V Southern Surveyor* and the Bio-optics voyage participants, particularly Massimo Pernice (Instituto de Ciencias del Mar- Consejo Superior de Investigacion Cientifica, Spain) for assistance with sampling, Andrew Bowie for dissolved Fe flow injection analyses, as well as Jennifer Clark (University of Technology Sydney) for flow cytometry analyses.

### Funding

This research was supported under the Australian Research Council's Discovery Projects scheme (DP1092892 to CH and MD; DP140101340 to MD; DP120102764 to JRS and MVB and FT130100218 to JRS) and the Marine National Facility. While doing this work, CH and KP were supported by a UTS Chancellor's fellowship, and JE by the NSW Science Leveraging Fund. The funders had no role in study design, data collection and analysis, decision to publish, or preparation of the manuscript.

### Grant Disclosures

The following grant information was disclosed by the authors:
Australian Research Council's Discovery Projects scheme: DP1092892, DP140101340, DP120102764, FT130100218.
Marine National Facility.
UTS Chancellor's fellowship.
NSW Science Leveraging Fund.

### Competing Interests

The authors declare there are no competing interests.

### Author Contributions

- Martina A. Doblin and Katherina Petrou conceived and designed the experiments, performed the experiments, analyzed the data, wrote the paper, prepared figures and/or tables.
- Sutinee Sinutok and Lauren F. Messer analyzed the data, prepared figures and/or tables and reviewed drafts of the paper.
- Justin R. Seymour performed the experiments, analyzed the data, reviewed drafts of the paper.
- Mark V. Brown performed the experiments, analyzed the data.
- Louiza Norman analyzed the data.
- Jason D. Everett and Allison S. McInnes analyzed the data, reviewed drafts of the paper.
- Peter J. Ralph contributed reagents/materials/analysis tools, reviewed drafts of the paper.
- Peter A. Thompson contributed reagents/materials/analysis tools.
- Christel S. Hassler conceived and designed the experiments, reviewed drafts of the paper.

## Data Availability

Accession numbers for sequence data in the NCBI Sequence Read Archive are SRR3275263 (nifH) and SRR3275264 (16S). All the other data from the study are available on the University of Technology Sydney data repository and data is provided as Supplemental Information.

## Supplemental Information

Supplemental information for this article can be found online at http://dx.doi.org/10.7717/peerj.1973#supplemental-information.

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
