# Peer review of "Nutrient uplift in a cyclonic eddy increases diversity, primary productivity and iron demand of microbial communities relative to a western boundary current"

_PeerJ, doi:10.7717/peerj.1973_

## Round 0.1 · original submission · Major Revisions

Please be sure to address all of the comments from the reviewers. Both reviewers note that the manuscript is hard to follow, so after making changes I would recommend sharing the draft with colleagues who are not marine microbial ecologists for feedback on its clarity.

·

Basic reporting

The manuscript tackles a highly relevant question: the movement of nutrients and microorganisms in the ocean and its relationship with biogeochemical cycles and foodweb. However, I found some parts need to be reworked because they are not well structured and have some mistakes. These issues affect the overall manuscript since it results not easy to follow the story. For instance:
In lines 281-293, authors should invite the reader to look at the t=0 in the plots, not have to find throughout the entire figure.
Fig S2 is cited several times in the text and should be as a manuscript figure. Maybe replace Fig 6, since it does not provide clear information (well-defines groups).
I suggest replace Fig 6E and 6F by abundance bar-plots such as Fig S2. MDS only says how distant samples are among them, but says nothing about the abundances described in text.
In the results section "Time-dependent response...", Fig S3 should be cited, not Fig S2. However, I think that Fig 7A and 7B is enough to get the idea and this supplementary figure does not contribute so much to the story.
I must disagree with the discussion of the Prochlorococcus decrease/increase. Differences in Fig 5E/5F were not significant respect to the control. In addition, I did not understand the phrase "suggesting that it could compete effectively with other microbes for N, but not for other nutrients".
Paragraph in lines 184-187 should be moved to the discussion, not in methods.
Paragraph in lines 446-451 fits better in the introduction.
Fig 3 legend should define the exes and the numbered white lines.
In the Fig 4 legend, I suggest to define which groups are looked by measuring the different pigments.
The same scale on y-axis should be used in Fig 4A/4B and 5I/5J. It is only ok for 5G/5H because the big difference.
Figs 7A/7B are also a community-level response and should be together with 7G/7H.
I suggest to change "nutrient enrichment" for "nutrient amendment" to refer to experimental nutrient input, in order to avoid confusion with microbial enrichment (a term commonly used in environmental microbiology).
What the authors mean with "reliable supply/source of eddies"?
Use t, not T, for time in text and figures. T is temperature.
Accession numbers are missed: sequencing data should be submitted and publicly available.
Unify style: use either μM or μmol L-1, not both.
Use cursive when referring to the taxons: Proteobacteria (e.g. alpha, beta, gamma), Rhodobacteriaceae.
Use cursive for the "P" for p-values in parentheses.
I liked the discussion of the "diazotroph relative abundance and diversity".

Specific details:
L28, L61: Define "Iron (Fe)" the first time is named, not in L64.
L164: Define CC.
L175: 16S rRNA and NifH genes.
L191: Define SSC
L214: nifH encodes a subunit, not the entire nitrogenase.
L221: an additional 10-cycles PCR
L227: I did not understand what authors mean in the parenthesis.
L323: Hex-Fuco
L426: Table 1
L458: Fig. 4
L494: prevous study (only one is cited).

Experimental design

I think the experimental setup is correct to assess the nutrient limitation. I only have a concern about the temperature used: it was the same of the sampling points (~21°C)?
If the temperature was the same as the sampling point, the conclusions are ok. In the case the temperature was different to the sampling point, it should be clearly stated in methods and discussed (specially in the Prochlorococcus discussion in L472).

Validity of the findings

I think the findings are ok. However I have some questions:
Shannon's index (L292) was calculated in base to OTUs or other?
How the authors explain the differences between t0 and control? manipulation during the experiments?

Additional comments

I feel the manuscrpt can be improved. The science is ok, but the writing is against the clarity of the story.

Reviewer 2 ·

Basic reporting

Table 1, 2, and 3: Can you add the full name of the abbreviations to the table text? This may mostly be an issue for people that aren’t specialists in the field, but TDFe from table 1, for example, is not described using this abbreviation elsewhere in the text (it’s referred to as DFe instead).

Are the lower case a, b, and c’s over the bar charts in figures 4, 5, and 7 from ANOVA calculations? Can the nature of these letters be clarified in the text?

Figure 5
The title is “photo and heterotrophic responses…” but it isn’t necessarily clear what the heterotrophs are referring to-notionally they are some fraction of picoeukaryotes and total bacteria, but the figure title suggests that specific heterotrophs are described.

Figure 6
It may add more clarity if text describing the clustering is added, e.g., clustering of samples is via a Bray-Curtis similarity matrix for pigmentation, 16S, and nifH abundances.

Have the 16S and nifH data been submitted to a public repository such as EBI-SRA? This needs to be noted in the text.

Experimental design

No comments.

Validity of the findings

Line 223
It may be worth briefly mentioning why the 95% identity was used. Also, the Penton paper used a complete linkage approach to clustering, which is more restrictive in the percent identity allowed of clustered reads than the UCLUST (greedy clustering) approach, i.e., Any given read is within 5% identity of another read for the complete linkage approach, while reads are only guaranteed to be within 5% identity of the centroid read for UCLUST, and some reads may be higher than 5% difference from one another. To recreate the Penton results, you would want to approximate it with 97.5% identity with the greedy clustering of UCLUST or use a method that allows average linkage with 95% identity.

Were there multiple comparisons corrections for the ANOVA results? In addition to the equal variance that was accounted for the text, there is also an assumption about normality of data. It may be safer to use a non-parametric Kruskal-Wallis test.

Additional comments

The addition of independent measurements of microbial shifts, such as flow cytometry, to the 16S sequencing results, is a welcome addition that is often lacking in microbial studies.

It may be useful to reiterate the relationship between climate change, WBC strength, and eddie formation/number from the Matear article in the conclusion to drive home the significance of the eddie research.

The text is well-written, but may be difficult for a non-marine microbial ecologist to follow.

Line 397-I think the appropriate figure reference is figure 7f.

Minor grammar issues:
Line 64 change “high” to “highly” or possible remove word.

Figure is sometimes written out as “figure”, other times as “fig.”

·

Basic reporting

The manuscript clearly reported the experiment.

Experimental design

I didn't think there were any major problems with the overall experimental design.

Validity of the findings

See below.

-- There were some points of interpretation that may need to be altered or better explained.

-- Some microbiological results (per-OTU differences) were not adequately statistically tested.

-- Are the microbiological data and metadata (especially the 16S rRNA data) deposited anywhere? I wasn't able to find this in the MS, but this point is important for later reuse/reanalysis.

Additional comments

While oceans can appear homogenous, they contain numerous features ranging from thin layers to meso-scale eddies that may influence their biogeochemistry and microbiology. Therefore, these features might influence calculations of carbon cycling important for climate models. Doblin and colleagues used a battery of techniques to investigate a cold-core eddy off the eastern coast of Australia, and compare microbial communities in the eddy to those in the current. Because cold-core eddies can upwell nutrients, the response of microorganisms to nutrient enrichment was addressed in mesocosm using nutrient enrichment experiments.

Some accommodations had to be made for the typical chaos of an oceanographic cruise (e.g. no trace metal free rosette, reuse of bottles between experiments, NO3 inadvertently present in Fe enrichment), but these limitations and the measures used to address them are transparently reported in the text, allowing for readers to form their own opinions. I didn’t personally find any of these hiccups to be a critical flaw preventing publication of the manuscript.

Overall, I thought this was an interesting and well-conducted study. There were some issues with some of the microbiological statistics that could be easily corrected, and there were a couple of points of interpretation that I think might either be problematic or ambiguous as presented.

Finally, there were several places where I would have liked to have seen a combined presentation of data from the EAC and the CCE so we could compare starting and ending microbial communities at each station in the same plot (ditto for the effects of each treatment on biomass production). I think this might allow for more direct testing of the author’s conclusions about whether these two water masses respond similarly or differently to nutrient enrichment.


In future work, I’d be very interested to see some of the microbial community comparisons conducted here replicated across several EAC/ CCE station pairs, in order to better explore whether there are consistent differences attributable to the cold-core eddy itself. However, I do understand that ship time is immensely expensive, and that it would likely be impractical to replicate the breadth of techniques applied here across many stations.

I suggest a minor revision, and discuss specific points in more detail below.

Major queries and comments:

Figure 6. I’m interested to know whether nutrient enrichment caused the EAC and CCE to become more similar in microbial community composition. If the main idea is that nutrient upwelling drives differences between the eddy and the current, should we expect that adding N to the EAC will move it closer to the control CCE conditions? Would it be possible to add a third column or supp. figure showing the two sample sets together on the same ordination plot to visualize whether this occurred?

Figure 7. G & H These figures show primary production as a function of nutrient enrichment in the eddy vs. the current. Since a key part of the take-away message seems to be the that nutrient limitation is different in the current vs. the eddy, would it be possible to directly test whether N enrichment had a significantly greater effect on primary productivity in the current relative to the eddy? A similar test for NFe enrichment would be great.

Table 2. Effect of experimental manipulation on microbial assemblages as shown by comparison of T72 results relative to T0. Since control vs. T0 represents bottle effects, shouldn’t other measurements be compared against the control rather than T0? For example we see strong negative differences in Prochlorococcus abundance in each treatment, but all are attributable to the strong negative difference in the EAC control. I’m wondering if this slightly different presentation might declutter the table substantially and let readers more rapidly hone in on the important (biological) differences. This might also help casual readers from drawing an unintended conclusion from the table.

>Line 262-264 The contribution of phytoplankton groups to the observed significant differences in community assemblage, as a function of treatment, were determined using Similarity Percentage Analysis (SIMPER; Clarke 1993).

I’d like to see per-taxon microbiological differences tested directly (e.g. with Kruskal-Wallis tests or your favorite parametric method) on the rarified OTU table (or taxon table), and corrected for multiple comparisons (e.g. with FDR). This would provide more rigorous statistical support for the significance of differences in particular bacterial groups than just reporting their contribution to significant overall community differences. These methods could also potentially help test significance for statements like those on lines 377-379 about OTUs that changed greatly in abundance in the nifH dataset despite a lack of significant difference in the overall community by ANOSIM.

Line 487: Addition of nitrate to the EAC community caused a doubling of total bacteria abundance within the first 24 h. Bacteria also became more abundant in the CCE community after 3 days within the Si and Mix bottles. Elevated bacteria abundance is likely to have arisen through increased dissolved organic matter (DOM) production by the resident community via several potential mechanisms: (1) mortality and cell lysis; (2) elevated rates of DOM release due to nutrient supplementation; and (3) altered microbial composition. While we did not measure rates of DOM production, our data certainly demonstrate coupling between autotrophs and heterotrophs,just as in previous studies (Baltar et al. 2010).

I follow the second two mechanisms, but I’m not sure I understand the first. In general, I get that bacterial cell lysis can contribute DOM to a system, and this DOM could then be partially recovered by other community members (e.g. during normal phage lysis of bacterial populations). But in a closed system like a nutrient enrichment bottle, I’m having trouble seeing how cell lysis could increase total bacterial counts, without invoking one of the other mechanisms. Certainly we don’t think overall biomass should increase in the closed system due to cell lysis, right? Is there some scenario for redistribution of biomass that is being implied, like replacement of cells that sequester a lot of carbon on a per cell basis with other strains that sequester less carbon per cell, resulting in greater overall cell counts? For the second mechanism, would it be worth adding a phrase clarifying that this is likely due to increased biomass production by phototrophs due to relief from nutrient limitation (if that is the intended meaning)?

Minor comments:

There are an awful lot of acronyms in the abstract. Are these necessary, or could the abstract be made more accessible without them? For example, is it important to remind the reader that it is the Eastern Australia Current (EAC) and a cold core eddy (CCE) every time the stations are mentioned, or could this information be introduced once, and then the sites referred to as ‘current’ and ‘eddy’ throughout? I don’t think this is a crisis either way, but mention it as a point to consider.

Figure S2. Do the gray bars represent unannotated sequences? I was having trouble finding gray on the color legend.

I found the introduction to be generally well-written and broadly accessible. However, it may be useful to define isopycnal surfaces (line 101) for environmental microbiologists without an oceanographic background. (I think a brief phrase or parenthetical note would be more than enough).

Line 107: It may be useful to mention what’s known about the host range of UCYN-A Cyanobacteria. Right now the text mentions that they are symbiotic, but doesn’t discuss their symbiotic partners. Again, I think a brief aside is all that’s needed. Would Hagiano et al PLoS One 2013 be appropriate here?

Line 113: “Using the reliable source of eddies in the EAC region…”

Suggest rephrasing slightly to clarify that the project drew on the known occurrence of eddies in the EAC region to test their effects (assuming that was the intended meaning). On a first read through it sounded like this was a reference to a known dataset (the ‘reliable source’) describing the coordinates of eddies in the EAC region. The meaning was clear after a moment, but I’m guessing this might slow down other readers.

>“The photosynthetic (PSC) and photoprotective (PPC) carotenoid pigment contributions were calculated as in Barlow et al. (2007), and the approach of Uitz et al. (2008) was used to assess the taxonomic composition of the phytoplankton community and characterise its size structure.”

Could this be expanded somewhat to describe in a bit more detail how pigment measurements were translated into phytoplankton classes, perhaps reprinting the relevant equation(s)? Also, while the authors note that the method has some limitations (due to overlapping size classes amongst plankton with a specific pigment etc) has this been quantified? Uitz 2008 shows correlation coefficients between the fraction of nano- pico- and micro- plankton with many physical and environmental parameters to often be quite low. Do we expect inferences of size classes of plankton from pigments to be similarly inaccurate, or is there some step in the prediction that gives better correlations than one might naively expect from the individual parameters? I don’t have any specific objection to this method, but think some of this information might help make it a little bit more clear exactly how the analysis was performed. In any case, I would expect this type of error to add noise to size class estimates but not bias them across treatments. Since in several cases significant differences were found (e.g. between nutrient-enrichment treatments) this seems like it wasn’t such a severe issue that it would preclude detecting trends.

Table 1. Despite having read the explanation in methods, It took me a moment on rereading this table to remind myself that the differences in sampling depth, which were substantial (40m vs. 80m) were due to the location of the chlorophyll a fluorescence maximum. Could a note be added to the table to remind readers why these depths differ?

---

## Round 0.2 · accepted · Accept

Please be sure to address the typographical issues noted by the reviewers. PeerJ does not perform copyediting, so it will be up to you to ensure that these changes are integrated.

·

Basic reporting

The manuscript was carefully reviewed and improved. I'm ok with the authors' responses. Now I feel it is more easy to follow the whole story.
Accession numbers should be the last requirement.
Finally, there are just 3 minimal details to correct before the printing process:
- Fig.4: In the png file the “μ” symbol is missing in the y-axis. Please check it.
- L243: nNifH...the "n" letter have to be removed.
- L347: Should say “Hex-Fuco”, not “Hexo-Fuco”.

Experimental design

No Comments.

Validity of the findings

No Comments.

Reviewer 2 ·

Basic reporting

A few minor issues that appear to have been introduced during revisions:
The text under figure 2B is a bit garbled, due to a circled 4:
“N* (4)M”
This also appears in Y-axis text of figure 4A/4B


Page 8:
Missing units (degrees C) on page 8 for deck board incubators statement.

Experimental design

No Comments

Validity of the findings

No Comments

Additional comments

I am generally satisfied with the authors' revisions.

Minor comments:
“foodweb” should be two words.

“prokaroyote” misspelled on page 10

Chlorophyll a sometimes listed as “Chl a” but most of the time as “Chl-a”

·

Basic reporting

No comments

Experimental design

No comments

Validity of the findings

No comments

Additional comments

Thanks for the opportunity to review this revised manuscript. The revision fully addressed all comments from the initial submission. Some important changes include depositing 16S data publicly; conducting additional statistical tests and ordination analyses; clarifying a few small but important points of interpretation in the text; and including additional information to make the manuscript accessible to a broad readership. I think the revised manuscript will be an important and useful contribution to its field, and recommend acceptance.